# Combining Deep Learning and Active Contours Opens The Way to Robust, Automated Analysis of 3D Brain Cytoarchitectonics

**Konstantin Thierbach**[*]
kthierbach@cbs.mpg.de

**Pierre-Louis Bazin**[†‡*]
p.bazin@nin.knaw.nl

**Walter de Back**[§]
walter.deback@tu-dresden.de

**Filippos Gavriilidis** [*]
gavriilidis@cbs.mpg.de

**Evgeniya Kirilina** [*¶]
kirilina@cbs.mpg.de

**Carsten Jäger** [*]
jaeger@cbs.mpg.de

**Markus Morawski** [‖]
markus.morawski@medizin.uni-leipzig.de

**Stefan Geyer** [*]
sgeyer@cbs.mpg.de

**Nikolaus Weiskopf** [**]
weiskopf@cbs.mpg.de

**Nico Scherf** [*]
nscherf@cbs.mpg.de

## Abstract

Deep learning has thoroughly changed the field of image analysis yielding impressive results whenever enough annotated data can be gathered. While partial annotation can be very fast, manual segmentation of 3D biological structures is tedious and error-prone. Additionally, high-level shape concepts such as topology or boundary smoothness are hard if not impossible to encode in Feedforward Neural Networks. Here we present a modular strategy for the accurate segmentation of neural cell bodies from light-sheet microscopy combining mixed-scale convolutional neural networks and topology-preserving geometric deformable models. We show that the network can be trained efficiently from simple cell centroid annotations, and that the final segmentation provides accurate cell detection and smooth segmentations that do not introduce further cell splitting or merging. The cell detection stage works sufficiently robust to even uncover actual errors in the reference annotations.

## 1 Introduction

Systematic studies of the cortical cytoarchitecture are indispensable to understand the functional organization of the human brain. Classical works based on qualitative description of cell counts and shapes in physical 2D sections of the human cortex revealed functional areas and segregation in the brain (Brodmann (1909); von Economo and Koskinas (1925); Vogt and Vogt (1919)). These

---

[*]Max Planck Institute for Human Cognitive and Brain Sciences, Leipzig, Germany

[†]Spinoza Centre for Neuroimaging, Amsterdam, The Netherlands

[‡]Netherlands Institute for Neuroscience Amsterdam, The Netherlands

[§]Institute for Medical Informatics and Biometry, Technische Universität Dresden, Dresden Germany

[¶]Center for Cognitive Neuroscience Berlin, Free University Berlin, Berlin, Germany

[‖]Paul Flechsig Institute of Brain Research, University of Leipzig, Leipzig, Germany

[**]The research leading to these results has received funding from the European Research Council under the European Union's Seventh Framework Programme (FP7/2007-2013) / ERC grant agreement n° 616905.

1st Conference on Medical Imaging with Deep Learning (MIDL 2018), Amsterdam, The Netherlands.

brain parcellations are currently updated and refined using automated image analysis (Zilles et al. (2002)). Even 3D imaging of post mortem brain tissue at microstructural resolution are within reach using recent light sheet fluorescence microscopy (LSFM) (Huisken et al. (2004); Dodt et al. (2007)) and tissue clearing protocols (Chung and Deisseroth (2013)). Combined with advanced image analysis these techniques enable studying cortical cellular organisation in the human brain with unsurpassed precision. Such studies are crucial to validate in vivo MRI-based cortical microstructure mapping (Weiskopf et al. (2015)) to understand the relationship between structure and function in the human brain. To reach this goal we need robust computational analysis relying on minimal manual annotations, facing the following challenges:

- Clearing of aged, unperfused human tissue is imperfect, and optical distortions due to scattering and refraction remain. This leads to varying background intensities across the image and shading artifacts.

- The penetration of antibody stains and thus contrast is uneven across the sample. The tissue degenerates with longer post-mortem times. These effects increase the already high variability of neural shape and appearance across the cortical samples.

- The dynamic range of intensities is large within and across cells (Fig.1a).

- The resolution is lower along the optical axis in the 3D stack. Additional imperfection in depth focusing and sample movement create artifacts through the depth of the stack (Fig.1b).

- Cell density varies locally, leading to false segmentation of cells into clusters.

Machine Learning methods improved the analysis of microscopy data (Sommer et al. (2011); Arganda-Carreras et al. (2017); Hilsenbeck et al. (2017)). Deep Learning, in particular Convolutional Neural Networks (CNNs), can address challenging problems in biomedical imaging because they learn multi-level internal representations of the data (LeCun et al. (2015); Shen et al. (2017)). These, typically supervised, methods require a lot of annotated data: For cell segmentation pixel-accurate masks have to be supplied (Ronneberger et al. (2015)). Manually annotating data for training is often prohibitive in biomedical applications where data are specialized, scarce and expert knowledge is required. Abstract concepts at the object level (Gestalt principles such as continuation, closure (Kandel et al. (2000)), or object topology) are hard to learn with CNNs. Additional annotation of the border region between adjacent cells is needed to reduce false merging of neighboring cells (Ronneberger et al. (2015)). Human vision exploits high level concepts using top-down processing (Kandel et al. (2000)) which is not represented in feedforward architectures.

Active Contour methods have been designed to embody high level concepts of object shapes. They can guarantee the smoothness of contours and a consistent topology (Bogovic et al. (2013); Han et al. (2003)): features that improve cell segmentation in challenging conditions and prevent splitting and merging of contours during segmentation. But active contour methods require an initialization with the number and approximate position of objects in the image. Robust initial localization of cells is hard to define a priori and should be learned from data. This is where Deep Learning has a clear advantage: CNNs can be trained to robustly predict cell positions in images using only sparse centroid annotations (Xie et al. (2016)).

We combine the complementary strengths of CNNs and topology-aware active contours into a robust workflow to detect and segment cells that delivers high quality results and requires only minimal annotations (sparse annotations of approximate cell centers are enough). Here, we present an extension of our previous 2D approach (Thierbach et al. (2018)) to 3D using data from microscopy image volumes obtained of cleared post mortem human brain blocks.

## 2 Methodology

### 2.1 Sample Preparation

Blocks from a human post mortem brain (temporal lobe cortex, male, 54 yr., post-mortem interval 96h) have been provided by the Brain Banking Centre Leipzig of the German Brain-Net. The entire procedure of case recruitment, acquisition of the patient's personal data, the protocols and the informed consent forms, performing the autopsy, and handling the autoptic material have been approved by the local ethics committee. For details on tissue preparation and clearing see (Morawski et al. (2017)).

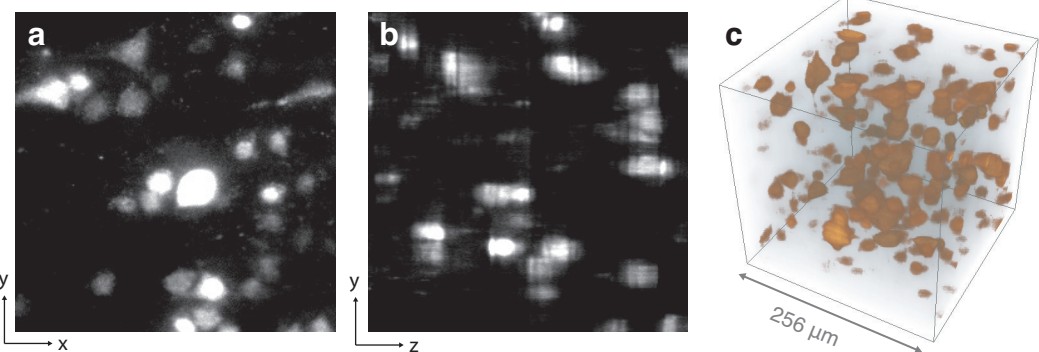

Figure 1: **Example of image data.** An xy (a) and yz (b) slice of a cuboid subvolume with side length of 256 $\mu m$ randomly sampled from the image stack. (c) Direct volume rendering of the subvolume.

## 2.2 Image Data

A commercial light-sheet fluorescence microscope (LaVision BioTec, Bielefeld, Germany) was used to image the cleared specimen. The microscope was equipped with 10x CLARITY-objective (Olympus XLPLN10XSVMP, numerical aperture (NA) 0.6, working distance (WD) 8 mm; Zeiss Clr Plan-Apochromat, NA 0.5, WD 3.7 mm) and operated with 630 nm excitation wavelength and band-pass 680 nm emission filter. Samples were stained with a fluorescent monoclonal antibody against human neuronal protein HuC/HuD (a specific marker for neuronal cells). The acquisition covered a 1.1 mm x 1.3 mm x 2.5 mm volume resulting in a stack of 2601 16 bit TIFF images (2560 x 2160 pixels, 0.51 $\mu m$ lateral resolution) using a 1 $\mu m$ step size.

For the 3D analysis pipeline we first resampled the image stack to an isotropic resolution of 1 $\mu m$. One expert manually annotated cell centroids for a subregion of the size 2304x256x1280 (z,x,y), which we subsequently used for the training of the convolutional neural network for cell localization. We additionally annotated cell centroids in three separate regions of size 256x256x256, which served as a test set to measure the performance of our method.

## 2.3 Cell Segmentation Workflow

The proposed method is an extension of the 2D approach described in Thierbach et al. (2018), based on a Fully Convolutional Neural Network (FCNN) for cell detection and a topology-preserving multi-contour segmentation (Bogovic et al. (2013)) to control smoothness and topology of the segmentation. In contrast to Thierbach et al. (2018) we use the Mixed-scale Dense Network (MS-D) architecture by Pelt and Sethian (2018) to directly predict masks of cell centroid regions in 3D. This step turned out to be more robust in 3D compared to regressing the Gaussian-smoothed centroid positions as proposed in Xie et al. (2016). An additional advantage of the MS-D architecture is the smaller number of parameters to reduce overfitting and the intrinsic parallel, multi-scale processing without resolution bottlenecks and upsampling steps. The basic concept of our approach is as follows:

- **Training step:** Pairs of image stacks (annotated centroids and raw data) are fed into MS-D network. The network is trained to directly segment a spherical region of radius 3 (voxels) around the annotated cell centroid.

- **Prediction step:** MS-D predicts probability maps of cell positions from the raw image. These centroid probabilities are thresholded at 90% probability level to derive cell positions to initialize the active contour segmentation that segments the cells from the raw images.

### 2.3.1 Cell Detection

We adapted the MS-D architecture by Pelt and Sethian (2018), with a *width* of 8 (corresponding to feature channels at each layer), a *depth* of 5 (corresponding to the spatial scale of receptive field sizes) and a kernel size of 5x5x5. We trained the network on pairs of randomly sampled image sections of size 96x96x96 pixels and binary reference images with the annotated centroids. The centroid stacks

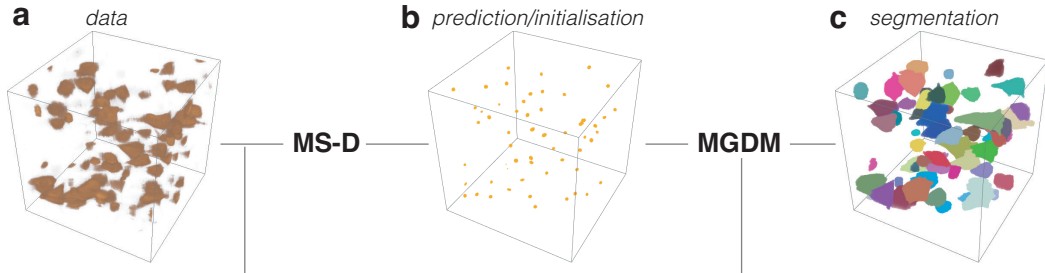

Figure 2: **Schematic overview of method.** The predicted cell positions are used as initialization and topology prior for the multi-object contour segmentation.

were convolved with a spherical kernel of size 3x3x3, to force the model to learn a segmentation of the cell center region. Segmenting only these small regions alleviated the problem of touching objects, and the prediction results can be used to locate the position of cells. In our experience, that leads to better results for 3D images than the regression approach from Thierbach et al. (2018). Further, we discarded any image sections which did not contain any annotated cells, to improve convergence. As loss function we choose $1 - F_\beta$ on the basis of binary pixel labels with $\beta = 0.7$ and

$$F_\beta = (1 + \beta^2) * \frac{precision * recall}{(\beta^2 * precision) + recall} \qquad (1)$$

With the choice of $\beta = 0.7$ we weighted the precision higher than the recall, laying a higher focus on correctly detecting cell centroid regions. For optimization we used stochastic gradient descent with a batch size of 2 distributed among two GeForce GTX 1080Ti, and an adaptive learning rate (ADADELTA) Zeiler (2012). To derive cell centroids from the MS-D predictions, we thresholded the prediction masks at 0.9.

### 2.3.2 Multi-Object Geometric Deformable Model

Once cell centroids have been detected, the final segmentation is handled by a Multi-object Geometric Deformable Model (MGDM) which ensures fast segmentation of an arbitrarily large number of cells while enforcing topological constraints (Bogovic et al. (2013)). The deformable model is driven by curvature regularization and balloon forces derived from the microscopy image intensities as follows.

For each detected cell, we first find the maximum intensity $M_c$ inside the initial centroid probability map. We set the balloon forces to decrease linearly with the distance to $M_c$:

$$F_c(x) = \frac{M_c - |I(x) - M_c|}{M_c}. \qquad (2)$$

where $I(x)$ is the image intensity. Because fluorescence intensity varies between cells, this calibration ensures that each cell is within its detection range. For the background, we first estimate the mean image intensity $M_B$ to separate background from cells and derive a similar balloon force:

$$F_B(x) = \frac{2M_B - I(x)}{M_B} \qquad (3)$$

To avoid unstable evolution due to large forces, $F_c$ and $F_B$ are all bounded in $[+1, -1]$. Balloon and curvature forces are combined in the MGDM evolution equation:

$$\frac{\partial \phi_{c,B}}{\partial t} = (w_\kappa \kappa + w_{c,B} F_{c,B}) |\nabla \phi_{c,B}| \qquad (4)$$

For this study we fixed the weights for curvature regularization to $w_\kappa = 0.1$, and the balloon forces to $w_{c,B} = 0.7$. The evolution was run for 200 iterations.

Note that the MGDM model uses a constant number of functions independently of the number of objects, and that object forces are simple functions of the image intensity, so memory requirements are independent of the number of detected cells, which can be quite important in the 3D case. Likewise, computational demands increase with the overall size of the boundary between objects, and a narrow band implementation ensures good scalability.

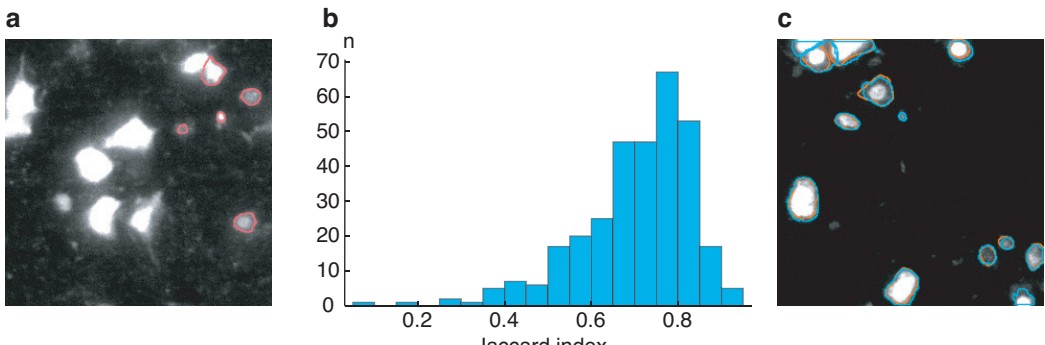

Figure 3: **Segmentation results**: The outlines on a slice from the test set in (a) show examples of detected objects that were not annotated in the reference data. The accuracy of segmentation masks is visualized as the histogram of the Jaccard index over the annotated masks across the entire test set. (c) An example plane from the test set showing the outlines of annotated reference masks (red) and segmentation results (blue).

### 2.3.3 Validation of Results

To assess cell detection accuracy, we computed precision $p = \frac{TP}{TP+FP}$, recall $r = \frac{TP}{TP+FN}$, and the combined F-score=$2\frac{p*r}{p+r}$ for the annotated image planes; $TP$ are true positive, $FP$ false positive, and $FN$ false negative detections. As manually segmenting cells in 3D is very laborious and error-prone we only annotated 2D reference cell masks in regularly spaced xy, and yz planes of the test images. To validate the agreement between annotated masks and segmentation we computed the segmentation entirely in 3D but exported the results only in those 2D planes that have been used for annotation. We then computed the Jaccard index between each reference cell mask $A$ and the best-fitting mask $B$ of the result in 2D: $J(A, B) = \frac{|A\cap B|}{|A|+|B|-|A\cap B|}$. As the segmentation is done in 3D we still get an appropriate estimate of the actual 3D segmentation performance using only the 2D reference masks.

## 3   Results

Examples of final segmentation result on a test sample are shown in Fig.2c and on a 2D sub-slice in Fig.3c (blue contours). The segmentation performed well across regions with varying cell appearance and density.

Quantitative results were aggregated over three test stacks and summarized in the following Table 3.

| precision | recall | F1-score | Jaccard index median |
|---|---|---|---|
| 0.810 | 0.873 | 0.840 | 0.732 |

The proposed method showed an overall good performance in localizing cells. However, in particular the precision was lower than expected from our previous 2D results (Thierbach et al. (2018)). To investigate if the false positive predictions can be systematically explained, we plotted the false positive cells over the raw images. It turned out that many of those predictions actually corresponded to cells (often with a low intensity) that were not correctly annotated in the reference data by the expert. See Fig.3 for an example. About 60% of the false positive detections were actually correct. Thus, our reported precision and F1-score is a conservative estimate of the actual performance.

The segmentation accuracy was very good throughout the test set. For a more detailed picture the distribution of the Jaccard index over all 321 masks is shown in Fig.3b. The MGDM segmentation tended to segment larger 3D masks compared to the manual reference as illustrated in Fig.3c showing the annotated outlines in red and the segmentation result in blue.

# 4  Conclusions

As a proof of concept we present a hybrid strategy to segment neural cells in 3D combining deep learning and a topology-preserving geometric deformable model. Our method robustly detects and segments cell bodies in light-sheet microscopy images of cleared post mortem human brain tissue. High-quality results were obtained despite large variations in cell shape and intensity, anisotropic resolution and challenging imaging artifacts. Our method only requires sparse annotation for training. This is a prerequisite for large-scale histological analysis of desired quality as fully annotated cell segmentations are very tedious and error-prone in 3D. As our basic concept is to learn cell characteristics from data to inform a segmentation model based on generic principles (e.g. smoothness of contours) it can be easily adapted to alternative clearing, staining and imaging protocols in the future.

The issue of correctly detected cells that were not annotated by the human expert illustrates two points: Firstly, correct manual annotation of 3D reference data is an error-prone process, in particular when cell intensities and shape varies in a dataset. One option to address this issues is collaborative annotation, where the inter- (and even the intra-) rater reproducibility can be taken into account, when comparing against a manual reference. Another option would be a two-step process for testing: an independent rater could classify the network predictions that did not match the reference as cells or errors. Secondly, these findings highlight that the MS-D network robustly learns structures of interest even from non-perfect reference annotations.

As a next step we will systematically optimize the neural network architecture, the loss function, and the forces of the MGDM segmentation to improve cell localization and segmentation further. Another interesting advance would be to train ensembles of networks to take the inter-network variability of predictions into account for downstream processing.

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
