# OpenReview forum: "Combining Deep Learning and Active Contours Opens The Way to Robust, Automated Analysis of 3D Brain Cytoarchitectonics"
_MIDL.amsterdam/2018/Conference — Submitted to MIDL 2018_

### Review · AnonReviewer1 · 2018-05-04
**Interesting application but paper could do with some additional re-organizing**

**Rating:** 2
**Confidence:** 2

**Review:**

Interesting hybrid method that combines a convolutional neural network with an active contour based method to locate and segment. Manuscript does a good job outlining the challenge and the need for automated segmentation. As I am not very familiar with this field it is hard for me to assess the originality and significance of the work. However, there are several references in the manuscript to a previous paper and based on the title of that paper this manuscript appears to describe some incremental changes to the previously described method.

Some issues that decrease my enthusiasm for the manuscript:
- Trained and tested on the data from a single subject (although I'm sure brain samples are hard to come by!)
- Algorithm and methods not very clearly described
- The reference standard used by the authors does not appear to be of very high quality

Detailed feedback:

2.2 Image Data
This only describes the collection and annotation of the training data for the cell localization. Later on the authors reference a set of data used to assess the segmentations. Is that the same data? Who set the reference standard for that data?

2.3 Cell Segmentation Workflow
The use of the term segmentation with respect to the centroid prediction is very confusing. Using localization would be better with the term segmentation describing the process as performed by the MGDM.

2.3.1 Cell Detection
Other than the last sentence of this section this is basically a description of the training process. It would be good to make that explicit.

2.3.2 Question for discussion: Does your method lose any cells localized in the first step due to segmentation errors by the MGDM?

2.3.3 Description of the data and reference standard for segmentation evaluation should be moved to the Image Data section.

3 Results
Who determine that 60% of FP were actually TP? The expert who set the reference standard? How many of the TP were actually FP?

4 Conclusions
I think the authors have some good ideas on how to improve their reference standard.


**Special Issue:**

No

---

### Review · AnonReviewer2 · 2018-05-09
**Promising work in a preliminary state. Missing system details, explorations and evaluations.**

**Rating:** 3
**Confidence:** 2

**Review:**

Preliminary work which is an extension to an existing work. Methodology is not detailed sufficiently, and results are very preliminary – authors claim that they are not accurately ground truthed (which means they need a second round of expert annotations to be conducted); also – no comparisons to other schemes are provided. The network part of the system is not detailed or investigated.

Paper summary
Authors present a proof-of-concept work that combines classical Active Contours method with a  mixed-scale dense convolutional neural (Pelt &  Sethian 2017) for initialization. This is a 3D extension to an earlier article by the authors” (Deep Learning meets Topology-preserving Active Contours: towards scalable quantitative histology of cortical cytoarchitecture”) . The system itself is not presented – but rather referenced out to a set of already published works.

Evaluation of the system: Authors provide segmentation results from the entire system (combining network and AC components). Of interest would be to explore results of the DL network part of the system. For example: a threshold of 0.9 was used on the CNN output – for the centroid extraction. The recall and precision of the centroid extraction (as opposed to pixel recall) could have been specified to explore the effect of the first stage on the final Jaccard measure.
A comparison of the results to other works is missing.

Pros
-	Novel idea utilizing advanced new and interesting DL component

cons
-Method presentation not self-contained. All parts are referenced out to other works.
-Lack of comparison to other works
-Metrics calculated for selected slices (due to understandable difficulties in segmentation )




**Special Issue:**

No

---

### Review · AnonReviewer3 · 2018-05-09
**A well written paper on microscopy segmentation but weakened by the datasets used.**

**Rating:** 3
**Confidence:** 2

**Review:**

The authors describe a method combining deep learning and active contours in a 3D application to segment cells in microscopy images.  The motivation for the combination with active contours is well described and the paper is in general well written.  The method is an extension from a 2D algorithm already published by the authors, but there are several notable differences in the 3D version, making it at least relatively novel in that context.
The quantity and quality of data used is of some concern since it comes only from a single patient.  It is not clear to me what kind of variability might be expected in images from different patients and what the implications of this are in terms of overfitting to the individual dataset.  This could have a significant impact on the validity of the reported results.
It also appears that there is no validation set used in this work, in other words all parameters and architectures are tuned to obtain best possible results on the test set.  Again, this has an impact on the validity of the described results.
For the reasons provided above I would not reject the paper outright but recommend it only tentatively for acceptance.

**Special Issue:**

No

---

### Decision · Program_Chairs · 2018-05-15
**Paper60 Acceptance Decision**

Reject